# Immunotherapy-Associated Renal Dysfunction in Metastatic Cancer: An Emerging Challenge in Onco-Nephrology

**DOI:** 10.3390/cancers17132090

**Published:** 2025-06-23

**Authors:** Francesco Trevisani, Andrea Angioi, Michele Ghidini, Matteo Floris, Davide Izzo, Renato Maria Marsicano, Nerina Denaro, Gianluca Tomasello, Ornella Garrone

**Affiliations:** 1URI—Urological Research Institute, Department of Urology, Division of Experimental Oncology, IRCCS San Raffaele Hospital, Via Olgettina 60, 20132 Milan, Italy; trevisani.francesco@hsr.it; 2Department of Nephrology, Dialysis, and Transplantation, G. Brotzu Hospital, Piazzale Alessandro Ricchi 1, 09047 Cagliari, Italy; andrea.angioi@aob.it (A.A.); matteo.floris@aob.it (M.F.); 3Department of Medical Oncology, Fondazione IRCCS Ca’ Granda Ospedale Maggiore Policlinico, Via Francesco Sforza 28, 20122 Milan, Italy; michele.ghidini@policlinico.mi.it (M.G.); nerina.denaro@policlinico.mi.it (N.D.); ornella.garrone@policlinico.mi.it (O.G.); 4Postgraduate School of Medical Oncology, University of Milan, Via Festa del Perdono 7, 20122 Milan, Italy; davide.izzo@unimi.it (D.I.); renato.marsicano@unimi.it (R.M.M.); 5Oncology Unit, ASST Crema, Largo U. Dossena 2, 26013 Crema, Italy

**Keywords:** immune checkpoint inhibitors, acute kidney injury (AKI), acute kidney disease (AKD), chronic kidney disease (CKD), onconephrology, immunotherapy, cancer

## Abstract

This study examines the incidence and patterns of renal dysfunction related to immune checkpoint inhibitors (ICIs) in a consecutive cohort of 226 metastatic patients with various solid tumors treated with immunotherapy. The aim was to better characterize the impact of immune-related nephrotoxicity in this setting. Acute kidney injury (AKI) developed in approximately 20% of patients within 90 days of initiating immunotherapy, and persisted beyond 30 days in 7% of cases. These findings highlight that renal complications are a frequent yet often underestimated adverse effect of ICI therapy, warranting close monitoring and appropriate management.

## 1. Introduction

Over the past decade, immune checkpoint inhibitors (ICIs) have revolutionized cancer treatment, improving survival outcomes across multiple metastatic malignancies [1,2]. By targeting PD-1, PD-L1, or CTLA-4 pathways, ICIs restore cytotoxic T-cell activity and induce durable tumor responses in a subset of patients with advanced disease [3].

However, the increasing use of ICIs has brought renewed attention to immune-related adverse events (irAEs), among which renal complications remain underrecognized [4,5]. Acute tubulointerstitial nephritis (ATIN) represents the most common renal irAE, though cases of acute tubular injury, glomerulonephritis, and other glomerular diseases have also been reported [6,7]. The incidence of ICI-associated acute kidney injury (ICI-AKI) ranges from 2% to 5%, with higher risks observed in patients receiving concomitant nephrotoxic medications or those with preexisting chronic kidney disease [8,9].

Beyond AKI, the concept of acute kidney disease (AKD)—defined as renal dysfunction persisting for 7–90 days, either following AKI or occurring de novo—has emerged as a clinically significant entity [10,11]. It reflects a subacute trajectory of injury with critical prognostic implications and has been formally acknowledged by the “Kidney Disease: Improving Global Outcomes” (KDIGO) guidelines as a stage deserving of clinical attention [12]. AKD has been associated with delayed renal recovery, progression to chronic kidney disease (CKD), and increased mortality, particularly among cancer patients [13,14].

Despite these concerns, data on the incidence, duration, and clinical impact of AKD in patients treated with ICIs remain limited [15]. Persistent renal dysfunction may necessitate therapy interruption, potentially compromising oncologic outcomes [16]. However, most published data focus on AKI as a binary, short-term event, neglecting the subacute phase where renal function fails to fully recover without meeting CKD criteria. This diagnostic gap may result in an underestimation of kidney injury burden and missed opportunities for nephrological intervention [17].

This study investigated the incidence and patterns of ICI-related renal dysfunction in a consecutive cohort of 226 metastatic patients with solid tumors (including melanoma, renal, lung, urothelial, colorectal, head and neck, breast, biliary tract, skin, and mesothelioma) treated with immune checkpoint inhibitors, aiming to better characterize immune-related nephrotoxicity in this setting.

## 2. Materials and Methods

### 2.1. Study Design and Setting

We conducted a retrospective cohort study at a single tertiary care academic center (Fondazione IRCCS Ca’ Granda Ospedale Maggiore Policlinico, Milan, Italy) to evaluate the incidence and predictors of acute kidney disease (AKD) in adult patients with solid tumors treated with immune checkpoint inhibitors (ICIs), with or without concurrent chemotherapy. The study period spanned June 2017 through December 2023.

This study was conducted in accordance with the Declaration of Helsinki. Ethical approval was obtained from the local Institutional Review Board (IRB). Patient confidentiality was maintained, and informed consent was waived due to the study’s retrospective nature.

### 2.2. Participants

We screened all patients aged ≥18 years who began ICI therapy (anti-PD-1, anti–PD-L1 or anti–CTLA–4 agents) during the study period. Exclusion criteria included baseline end-stage kidney disease (eGFR < 15 mL/min/1.73 m^2^ or hemodialysis or peritoneal dialysis), incomplete critical clinical data (e.g., baseline renal function), pregnancy, and follow-up of less than 30 days after ICI initiation. A total of 226 patients satisfied the inclusion criteria (Table 1).

### 2.3. Data Collection

Demographic and clinical variables were extracted from electronic health records by trained researchers, including age, sex, body surface area (BSA), smoking history, comorbidities (hypertension, diabetes, chronic kidney disease, ischemic heart disease, chronic obstructive pulmonary disease [COPD]), concomitant medications (ACE inhibitors/ARBs, diuretics, anti-inflammatories, antibiotics), cancer type and stage, PD-L1 status, line of ICI therapy, and receipt of concurrent cytotoxic chemotherapy agents (taxanes, platinum compounds, targeted therapies). Laboratory data included baseline and on-treatment serum creatinine, blood counts (white cells, neutrophils, lymphocytes, eosinophils, platelets), and C-reactive protein. Baseline serum creatinine was determined as the most recent creatinine level recorded before the initiation of ICI therapy. Serum creatinine concentrations were collected longitudinally after the onset of AKD. The 2021 Chronic Kidney Disease Epidemiology Collaboration (CKD-EPI) equation was used to calculate the estimated GFR (eGFR) in mL/min/1.73 m^2^.

### 2.4. Definitions

AKD was defined according to KDIGO 2024 criteria as the onset of acute kidney injury (AKI) or eGFR < 60 mL/min/1.73 m^2^, or eGFR decrease by ≥35%, or serum creatinine increase by 50% occurring less than 90 days after a potential insult (in this case ICI initiation), which may precede and/or be superimposed on CKD [12].

### 2.5. Outcomes

The primary outcome was the development of acute kidney disease (AKD) after initiating immune checkpoint inhibitor (ICI) therapy.

### 2.6. Statistical Analysis

Continuous variables were summarized as medians with interquartile ranges and tested for normality using the Shapiro–Wilk test. Non-normally distributed variables were compared using the Mann–Whitney U test. Categorical variables were expressed as counts and percentages and compared with the Chi-square test or Fisher’s exact test as appropriate. Variables with *p* < 0.05 in univariate analysis were considered for multivariate modeling.

Using a backward stepwise selection approach, we constructed a multivariate logistic regression model to identify independent predictors of AKD. The delta eGFR worst value was excluded from the multivariate model to avoid circularity, as it is intrinsic to the AKD definition. Adjusted odds ratios (ORs) with 95% confidence intervals (CIs) were reported. Model discrimination was assessed using the area under the receiver-operating characteristic curve (AUC); calibration was evaluated with the Hosmer–Lemeshow goodness-of-fit test. Statistical analyses were performed using JASP (version 0.19.3; JASP Team, Amsterdam, The Netherlands). A *p*-value of <0.05 was considered statistically significant.

We used a complete case approach without imputation for missing laboratory values.

## 3. Results

### 3.1. Baseline Characteristics of the Study Population

The study cohort included 226 patients treated with ICIs for various malignancies. The patients’ characteristics are summarized in Table 1 and Table 2. The median age was 69 years (IQR, 62–75), and 37.2% of the population was male. The median body mass index (BMI) was 23.71 kg/m^2^ (IQR, 21.4–26.8), and the median body surface area (BSA) was 1.76 m^2^ (IQR, 1.60–1.92).

The population had a high prevalence of comorbid conditions, including hypertension in 42.0%, diabetes mellitus in 15.0%, ischemic heart disease in 10.6%, COPD in 10.2%, and CKD in 4.0% of cases. Prior viral infections were reported in 13.5% of patients.

Regarding smoking status, 18.6% were never smokers, 34.1% were former smokers, and 33.2% were current smokers. Among the 154 patients with pack year data available, the median smoking exposure was 35 pack years (IQR, 17.3–50.0).

Pharmacologic profiles at baseline revealed the widespread use of cardiovascular and metabolic agents. A total of 33.6% of patients were receiving either angiotensin-converting enzyme (ACE) inhibitors or angiotensin receptor blockers (ARBs), while 16.8% were on calcium channel blockers and 8.4% were taking diuretics. Antiplatelet therapy was reported in 19.9% of cases, and beta-blockers were used in 11.9%. Regarding antidiabetic treatment, 11.9% of patients were on oral antidiabetic agents, and 3.1% were receiving insulin. Anti-inflammatory drugs were documented in 2.7% of cases. Additionally, 14.8% of patients had been prescribed antibiotics at the time of treatment initiation.

The most common cancer type was non-small-cell lung adenocarcinoma (50.9%), followed by melanoma and renal cell carcinoma (each 7.5%), squamous cell lung cancer (8.4%), urothelial carcinoma (7.1%), mesothelioma (3.1%), breast cancer (3.1%), and other malignancies including small-cell lung cancer, melanoma, squamous cell carcinoma of the skin, colon, biliary tract, and head and neck cancers. Tumor invasion characteristics included vascular invasion in 5.8%, lymphatic invasion in 2.7%, and neural invasion in 4.4%.

The median time from diagnosis to treatment initiation was 68 days (IQR, 36.3–352.0).

Regarding treatment, 58.8% of patients received ICIs as first-line therapy, while 27.0%, 2.7%, 1.3%, and 2.7% received them as second, third, fourth, or fifth-line therapies, respectively. A total of 36.3% received combined chemotherapy, most frequently taxanes (10.6%) or platinum agents (9.7%). Targeted therapies were administered in 7.1%, radiotherapy in 3.5%, and immunotherapy alone in 1.3%.

ICI targets included PD-L1 in 73.5%, PD-1 in 21.7%, and CTLA-4 in 4.9% of cases. Tumor PD-L1 expression status was known in 42.0% of patients.

At baseline laboratory exams, white blood cell (WBC) counts had a median of 7.3 × 10^3^/mm^3^ (IQR, 5.92–8.73), with neutrophils at 4.49 × 10^3^/mm^3^ (IQR, 3.56–5.77) and lymphocytes at 1.635 × 10^3^/mm^3^ (IQR, 1.218–2.172). The neutrophil-to-lymphocyte ratio (NLR) was 2.67 (IQR, 1.80–4.16), and eosinophils were 0.15 × 10^3^/mm^3^ (IQR, 0.08–0.263). The eosinophil-to-lymphocyte ratio (ELR) and platelet-to-lymphocyte ratio (PLR) were 0.08 (IQR, 0.04–0.15) and 151.5 (IQR, 98.9–228.6), respectively. Baseline platelet count was 260 × 10^3^/mm^3^ (IQR, 202–308). C-reactive protein (CRP) levels were available in 133 patients, with a median of 0.49 mg/dL (IQR, 0.18–1.90).

The median eGFR was 79.07 mL/min/1.73 m^2^ (IQR, 61.15–92.35).

### 3.2. Renal Function Follow up and Oncological Outcomes

During ICI therapy, eGFR values remained relatively stable across treatment cycles, showing moderate fluctuations without substantial declines in the overall population. Specifically, median eGFR values following each treatment cycle ranged from 73.61 to 74.88 mL/min/1.73 m^2^, indicating the absence of significant cumulative renal impairment at the population level.

Nevertheless, a subset of patients experienced clinically meaningful deterioration in renal function. The median absolute decline in eGFR at the worst recorded time point (ΔGFR) was 9.71 mL/min/1.73 m^2^ (IQR, 3.74–17.09), corresponding to a median percentage reduction of 14.1% (IQR, 5.12–23.49). Acute kidney disease (AKD) occurred in 46 patients (20.4%), and among these, 16 patients (7.1%) developed persistent AKD lasting beyond 30 days. Furthermore, a significant decline in renal function, defined as a reduction in eGFR of at least 30% from baseline, was observed in 29 patients (12.8%).

Regarding the oncological outcome, 48.2% of patients died, and 61.4% experienced disease progression or recurrence. Treatment-related toxicity occurred in 49.6% of cases, with grade ≥2 adverse events in 47.8%. At the first follow-up, complete and partial remission were observed in 15.1% and 42.5% of patients, respectively. At second follow-up, complete remission was seen in only 0.4%, and partial response in 9.3%.

The median follow-up from the initiation of first-line therapy was 671 days (IQR, 366–1042), and it was 537 days (IQR, 299–850) from second-line initiation. Among patients with disease recurrence (*n* = 133), the median follow-up duration from recurrence was 189 days (IQR, 67–422). No patient with persistent AKD progressed to dialysis-dependent renal failure.

### 3.3. Univariate Analysis: Predictors of AKD

A univariate analysis was conducted to explore potential associations between clinical, pathological, treatment-related, and laboratory variables, and the development of AKD in patients receiving ICIs (Table 3).

Among the categorical variables, several factors were significantly associated with the occurrence of AKD. Firstly, the line of treatment showed a significant association (*p* = 0.033), indicating a higher risk in patients receiving ICIs beyond the first line. The use of concomitant chemotherapy was also related to AKD development (*p* = 0.030), possibly reflecting the additive renal burden of combined therapeutic strategies.

Among comorbidities and baseline medications, ischemic heart disease (*p* = 0.027), diuretic use (*p* = 0.014), anti-inflammatory drugs (*p* = 0.004), and antibiotics (*p* = 0.035) were all significantly associated with the onset of AKD. These findings suggest that both cardiovascular comorbidity and nephrotoxic or immunomodulatory agents may predispose patients to renal dysfunction during immunotherapy. Hypertension and platinum-based chemotherapy showed trends toward significance, further supporting a possible link between preexisting endothelial damage or cytotoxic exposure and renal vulnerability.

Analysis of continuous variables revealed additional predictors (Table 4). Patients who developed AKD had significantly higher body surface area (BSA) (*p* = 0.006), suggesting that greater drug distribution volume or exposure may contribute to renal risk. Eosinophil counts were also elevated in AKD patients (*p* = 0.018), aligning with hypotheses of immune-mediated renal injury. Moreover, platelet counts were significantly different between groups (*p* = 0.014), while the platelet-to-lymphocyte ratio showed a borderline association.

Serial renal function measurements during treatment revealed a clear trend: although baseline eGFR was not predictive of AKD, post-treatment values became progressively more indicative. Significant differences in eGFR were observed after cycles 1 through 3, with highly significant reductions emerging from cycle 4 onward (all *p* < 0.001). These findings suggest that AKD is not necessarily evident at baseline but may evolve gradually with cumulative exposure to ICIs or concurrent factors. Although regular urinalysis was performed in all patients, sediment findings were nonspecific and could not differentiate acute tubular injury from interstitial nephritis reliably.

No significant associations were found between AKD and overall survival, duration of follow-up, or response to treatment. The degree of AKD observed in our cohort was generally mild and managed without prolonged interruptions of immune checkpoint inhibitor therapy, thereby explaining the lack of a discernible effect on treatment continuity and overall survival within the duration of our study.

### 3.4. Multivariate Analysis: Predictors of AKD

A multivariate logistic regression model was performed to identify independent predictors of AKD in patients treated with immune checkpoint inhibitors (Table 5). Variables included in the model were selected based on both clinical relevance and statistical significance in the univariate analyses. The variable delta eGFR (worst value), although strongly associated with AKD, was excluded from the model to avoid collinearity, as it is intrinsically related to the AKD definition.

In the final model, body surface area (BSA) emerged as a statistically significant independent predictor of AKD. A higher BSA was associated with an increased risk (OR = 8.17, 95% CI: 1.23–54.0, *p* = 0.030), suggesting that individuals with larger body surface areas may be more susceptible to renal adverse events, possibly due to increased drug exposure or metabolic demand.

The use of anti-inflammatory drugs at baseline was also independently associated with a markedly elevated risk of AKD (OR = 29.74, 95% CI: 1.95–453.10, *p* = 0.014), reinforcing the known nephrotoxic potential of non-steroidal anti-inflammatory drugs (NSAIDs) in the context of immune-mediated renal stress.

A trend toward statistical significance was observed for several other variables. Patients receiving antibiotics at baseline had a higher likelihood of developing AKD (OR = 3.02, 95% CI: 0.74–12.29, *p* = 0.054), potentially reflecting the impact of microbiome disruption or pre-existing infection. Similarly, diuretic use showed a near-significant association with an increased AKD risk (OR = 2.97, *p* = 0.068), possibly due to volume depletion or prerenal hypoperfusion.

Baseline platelet count showed a borderline inverse association with AKD risk (OR = 0.996 per unit increase, *p* = 0.061), while baseline eosinophil count, although not significant, trended toward increased risk (OR = 1.38, *p* = 0.122), potentially reflecting an immunologic contribution to renal pathophysiology.

### 3.5. Model Performance

The logistic model showed acceptable discrimination (AUC, 0.778) and good calibration on the Hosmer–Lemeshow test (*p* > 0.05), with the ROC curve illustrated in Figure 1.

The ROC curve illustrates the discriminative ability of the multivariate logistic regression model derived through a stepwise approach for predicting AKD. The AUC is 0.778, indicating acceptable model performance.

## 4. Discussion

The advent of ICIs has markedly transformed the therapeutic landscape for multiple advanced malignancies, including melanoma, renal cell carcinoma, non-small-cell lung cancer (NSCLC), urothelial carcinoma, and others. By targeting regulatory pathways such as PD-1, PD-L1, and CTLA-4, ICIs restore antitumor T-cell activity, improving survival outcomes across several tumor types [1,18]. However, the expanding use of ICIs has been accompanied by a parallel increase in immune-related adverse events (irAEs), which may affect virtually any organ system, including the kidneys [19]. Among renal complications, acute kidney injury (AKI) has been the most extensively studied and recognized form of kidney damage in oncologic literature over the past decade due to its acute onset marked by a rapid decline in glomerular filtration rate (GFR), decreased urine output, and in severe cases leading to end-stage kidney disease (requiring dialysis), potentially leading to significant morbidity and oncological therapy discontinuation and interruption [19,20]). AKI-associated with ICIs requires prompt and systematic management to minimize complications. According to American Society of Clinical Oncology (ASCO) guidelines [21], key strategies include regular monitoring of serum creatinine (baseline, before each cycle, and as clinically indicated) and urine output, particularly during the first 3 months of therapy; urinalysis (to detect proteinuria, hematuria, or sterile pyuria) and assessment of electrolytes. ICI temporary suspension and corticosteroids (prednisone 0.5–1 mg/kg/day) are recommended for grade ≥2 AKI (KDIGO criteria). In more severe cases (grade 3–4), high-dose intravenous methylprednisolone (1–2 mg/kg/day) is indicated along with permanent ICI discontinuation. Kidney biopsy should be considered if diagnosis is unclear (e.g., no improvement with steroids) or in presence of atypical features (e.g., nephrotic-range proteinuria). Alternative immunosuppression with mycophenolate mofetil or infliximab should be administered in case of no response to steroids within 48–72 h. Supportive care including hydration, avoidance of nephrotoxic agents (e.g., NSAIDs, contrast dyes), and electrolyte management should always be offered. However, timely multidisciplinary collaboration involving nephrology specialists is crucial since early identification of AKI may help avoid treatment discontinuation and subsequent adverse effects on cancer outcomes.

The introduction of the AKD concept by the KDIGO guidelines in 2012 provided an important framework for characterizing subacute kidney injury that does not fully meet criteria for either AKI or CKD [22,23]. This classification has offered clinicians a valuable tool for identifying and managing this distinct clinical entity, particularly in oncology settings where early renal dysfunction recognition may significantly impact treatment decisions [11,24]. For all the above mentioned reasons, our study aimed to investigate the incidence, clinical characteristics, and risk factors for AKD in a real-world consecutive cohort of 226 metastatic cancer patients treated with ICIs.

Our findings demonstrated that AKD occurred in 20.4% of patients within 90 days of ICI initiation. In 7.1% of patients, it persisted beyond 30 days, suggesting that subacute renal dysfunction is a clinically relevant and underrecognized entity in the context of cancer immunotherapy. This incidence of AKD was notably higher than commonly reported rates of ICI-associated AKI, typically ranging from 2% to 5%, indicating that the use of AKI criteria alone may underestimate the burden of renal impairment in this setting [25,26].

In our study, baseline use of nonsteroidal anti-inflammatory drugs (NSAIDs) was the most significant predictor of AKD (OR 29.74, *p* = 0.014), consistent with the nephrotoxic potential of these agents and their capacity to impair renal autoregulation by inhibiting prostaglandin synthesis [27]. NSAID use has been implicated in increasing the risk of ICI-associated interstitial nephritis, particularly when combined with other nephrotoxins such as antibiotics [15]. Despite the small number of NSAID-exposed patients in our cohort (*n* = 6), the known nephrotoxic potential of these agents and our observed association support avoiding NSAIDs during ICI therapy, especially in the intensive early treatment phase. Consistent with current guidelines, regular monitoring of renal function, performed at minimum before each cycle, should always be guaranteed.

Additionally, we found that a higher body surface area (BSA) was an independent risk factor for AKD (OR 8.17, *p* = 0.03), suggesting that larger distribution volumes may alter pharmacokinetics and toxicity thresholds for ICIs or co-administered nephrotoxins. Elevated baseline eosinophil count was also associated with increased AKD risk (*p* = 0.018), possibly reflecting a predisposition to type IV hypersensitivity reactions and interstitial nephritis [28].

Although we identified several clinical and biochemical predictors of AKD, we did not observe a significant difference in the incidence of other grade ≥2 irAEs between patients who developed AKD and those who did not. This finding suggests that AKD may arise through distinct immunopathological mechanisms, potentially independent of systemic immune activation, or may reflect the subclinical nature of renal involvement, often overlooked due to the absence of overt symptoms and the slow, progressive course of injury.

Nevertheless, the prognostic impact of AKD should not be underestimated. In our cohort, nearly half of the patients (48.2%) died during follow-up and 61.4% experienced disease progression. These figures highlight that renal injury during immunotherapy is not merely an epiphenomenon but may significantly influence oncological outcomes. Prior studies have shown that even modest declines in estimated glomerular filtration rate (eGFR) correlate with inferior progression-free and overall survival, reinforcing the concept that renal function is not just a collateral parameter but a critical determinant of treatment continuity and long-term prognosis [12].

Whether AKD itself directly contributes to these poor outcomes or rather serves as a marker of underlying frailty, systemic inflammation, or cumulative comorbidity remains an open question. Future prospective studies are warranted to investigate causality and explore whether early detection and nephroprotective interventions could mitigate both renal and oncologic deterioration.

Of particular concern was the persistence of AKD beyond 30 days in one-third of affected patients. Persistent renal dysfunction has been shown to increase the risk of CKD and is independently associated with cardiovascular morbidity and all-cause mortality [29]. As such, transient monitoring strategies that fail to follow renal function over time may miss the window for early intervention. In our previous works, we have highlighted that AKD is often overlooked in patients who do not meet AKI criteria but still exhibit meaningful renal impairment that may alter long-term outcomes [24,30].

Furthermore, the role of eosinophils as a biomarker of immune-mediated renal injury warrants further investigation. Eosinophil-rich infiltrates are frequently observed in cases of acute interstitial nephritis (AIN), which is the predominant histopathological finding in ICI-related AKD [6]. As such, pre-treatment eosinophilia may serve as a risk marker, prompting closer renal monitoring or preventive measures in high-risk patients.

Another critical finding was the association between early decline in eGFR during initial ICI cycles and subsequent AKD development. This highlights the importance of dynamic renal function assessment, rather than relying solely on baseline values. In some patients, early eGFR decline may reflect subclinical tubular injury or evolving interstitial inflammation, which could be amenable to early corticosteroid intervention if recognized promptly [19]. Moreover, it is possible that in such a cohort of patients, the value of serum creatinine was not reliable, and therefore, the total GFR was less than expected, making these patients more fragile in terms of renal dysfunction.

Another important aspect is related to the high incidence of AKD among patients with baseline ischemic heart disease and diabetes. This aligns with the concept that “multi-hit” injury—where ICIs act as an immune trigger on a background of preexisting renal susceptibility—underlies most renal irAEs. Therefore, risk stratification tools incorporating clinical, biochemical, and therapeutic variables are urgently needed to predict AKD and guide pre-treatment counseling and monitoring strategies.

Based on our findings, we propose that from a pharmacovigilance standpoint, AKD should be formally recognized as a reportable renal event in future clinical trials and safety registries involving immune checkpoint inhibitors. Existing pharmacovigilance frameworks typically focus on AKI using short-term criteria, thereby failing to capture a substantial proportion of patients with persistent renal dysfunction that does not meet AKI thresholds. Incorporating AKD into safety monitoring protocols would enable earlier identification of subclinical renal toxicity and more accurately reflect the burden of nephrotoxicity associated with ICIs. Furthermore, longitudinal tracking of renal function trajectories over time would allow for improved risk stratification, a better understanding of delayed-onset toxicities, and refinement of dose–toxicity relationships.

Finally, our study has several limitations. (1) Kidney biopsies were not performed in our cohort; consequently, all cases of AKD were diagnosed clinically based on observed serum creatinine trends and corroborative laboratory parameters. (2) This is a single-center cohort, and institution-specific protocols and practices may influence our findings; therefore, external validation in diverse clinical settings remains essential. (3) Although concurrent chemotherapy showed a non-significant protective trend (*p* = 0.086), small subgroup sizes precluded detailed analysis of differential renal impacts of platinum versus taxane-based regimens, underscoring the need for further investigation in larger cohorts. (4) Some independent variables, like NSAIDs at baseline, had only six patients, potentially inflating the observed OR of 29.74, requiring larger cohorts. (5) Our study did not systematically evaluate long-term CKD progression or dialysis requirements. (6) Prospective studies systematically characterizing irAE timing, severity, and biomarkers could significantly enhance mechanistic understanding and clinical management of AKD associated with ICIs. (7) Despite our efforts to apply consistent AKD definitions and adjust for ICI class and treatment line, residual temporal confounding remains possible due to evolving practice patterns and diagnostic awareness. (8) Baseline serum creatinine was taken as the most recent value within 30 days before ICI therapy; potential pre-treatment variability in cancer patients may lead to misclassification of AKD onset.

These findings highlight the need for prospective multicenter validation studies incorporating standardized AKI definitions, investigation of non-invasive blood or urinary biomarkers for early detection, and advanced imaging modalities to better characterize renal pathology. Such efforts could establish more sensitive diagnostic criteria and guide optimal monitoring strategies without compromising cancer treatment efficacy.

## 5. Conclusions

In conclusion, our study demonstrates that AKD is a common and clinically significant complication in patients with metastatic solid tumors treated with ICIs. Recognizing AKD as a discrete entity, distinct from AKI and CKD, allows for improved risk stratification and targeted management. Given the increasing use of immunotherapy in oncology, integrating renal surveillance into routine care pathways is essential to preserve renal function and treatment continuity.

## Figures and Tables

**Figure 1 cancers-17-02090-f001:**
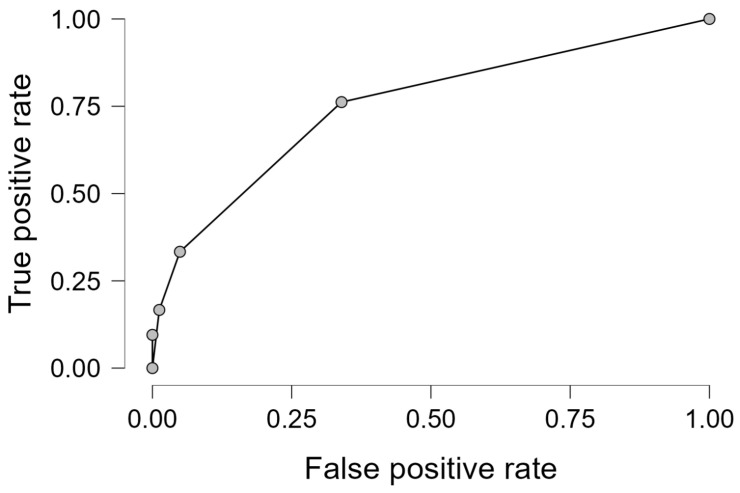
ROC for the final logistic regression model.

**Table 1 cancers-17-02090-t001:** Descriptive statistics of the cohort (categorical variables).

Category	Variable	Frequency	Percentage (%)
Sex	Male	84	37.2
Risk factors	Viral infections	30	13.5
	Diabetes	34	15.0
	Hypertension	95	42.0
	Ischemic heart disease	24	10.6
	COPD	23	10.2
	Chronic kidney disease	9	4.0
Smoking status	Never	42	18.6
	Former	77	34.1
	Current	75	33.2
Cancer characteristics	PD-L1 status	95	42.0
	Vascular invasion	13	5.8
	Lymphatic invasion	6	2.7
	Neural invasion	10	4.4
Other drugs	Antibiotics	31	14.8
	ACE inhibitors/ARBs	76	33.6
	Calcium channel blockers	38	16.8
	Diuretics	19	8.4
	Oral antidiabetics	27	11.9
	Insulin	7	3.1
	Anti-inflammatories	6	2.7
	Antiplatelet agents	45	19.9
	Beta-blockers	27	11.9
Anti-cancer treatment	Associated chemotherapy	82	36.3
	Taxanes *	24	10.6
	Platinum **	22	9.7
	Targeted therapy ***	16	7.1
	Immunotherapy	3	1.3
	Chemotherapy	29	12.8
	Radiotherapy	8	3.5
Cancer type	Melanoma	17	7.5
	Kidney	17	7.5
	NSCLC	115	50.9
	Squamous lung	19	8.4
	Urothelial	16	7.1
	Skin	9	4.0
	Colon	2	0.9
	Head and neck	9	4.0
	SCLC	4	1.8
	Mesothelioma	7	3.1
	Biliary tract	4	1.8
	Breast	7	3.1
ICI target	PD-1	166	73.5
	PD-L1 status	49	21.7
	CTLA4	11	4.9
Treatment line	1st	133	58.8
	2nd	61	27.0
	3rd	6	2.7
	4th	3	1.3
	≥5th	6	2.7
Kidney outcomes	AKD flag	46	20.4
	Persistent AKD	16	7.1
	eGFR loss ≥30%	29	12.8
Overall outcomes	Death	109	48.2
Chemotherapy outcomes	Toxicity	112	49.6
	Toxicity grade ≥2	108	47.8
	Progression/Recurrence	132	61.4
Best cancer response	Complete remission (1st FU)	33	15.1
	Partial remission (1st FU)	93	42.5
	Complete remission (2nd FU)	1	0.4
	Partial remission (2nd FU)	21	9.3

* paclitaxel, docetaxel. ** carboplatin, cisplatin. *** cabozantinib, entrectinib, osimertinib, sunitinib, cetuximab.

**Table 2 cancers-17-02090-t002:** Descriptive statistics of the cohort (continuous variables).

Variable	Valid	Median	IQR	Shapiro–Wilk	*p*-Value	25th Percentile	75th Percentile
Age (years)	226	69.000	13.000	0.966	<0.001	62.000	75.000
Diagnosis to treatment time (days)	226	68.000	315.750	0.415	<0.001	36.250	352.000
Pack/Year (Tobacco)	154	35.000	32.750	0.850	<0.001	17.250	50.000
Body mass index (BMI)	224	23.710	5.400	0.880	<0.001	21.400	26.800
Body surface area (BSA)(m^2^)	223	1.760	0.320	0.987	0.043	1.600	1.920
WBC (×1.000/mm^3^)	225	7.300	2.810	0.607	<0.001	5.920	8.730
Neutrophils (×1.000/mm^3^)	225	4.490	2.210	0.704	<0.001	3.560	5.770
Lymphocytes (×1.000/mm^3^)	224	1.635	0.955	0.231	<0.001	1.218	2.172
N/L ratio	224	2.670	2.363	0.745	<0.001	1.799	4.162
Eosinophils (×1.000/mm^3^)	224	0.150	0.182	0.255	<0.001	0.080	0.263
E/L ratio	224	0.080	0.111	0.179	<0.001	0.040	0.151
Platelets (×1.000/mm^3^)	225	260.0	106.0	0.908	<0.001	202.0	308.0
P/L ratio	224	151.464	129.762	0.887	<0.001	98.867	228.630
CRP (basal) (mg/dL)	133	0.490	1.720	0.418	<0.001	0.180	1.900
eGFR (basal) (mL/min/1.73 m^2^)	225	79.066	31.198	0.986	0.030	61.151	92.350
eGFR after cycle 1 (mL/min/1.73 m^2^)	224	73.605	30.058	0.995	0.621	60.460	90.518
eGFR after cycle 2 (mL/min/1.73 m^2^)	225	74.258	31.676	0.992	0.255	58.236	89.912
eGFR after cycle 3 (mL/min/1.73 m^2^)	209	74.025	31.327	0.992	0.295	58.780	90.107
eGFR after cycle 4 (mL/min/1.73 m^2^)	198	74.185	31.510	0.988	0.086	58.155	89.665
eGFR after cycle 5 (mL/min/1.73 m^2^)	181	73.644	28.384	0.993	0.520	58.775	87.159
eGFR after cycle 6 (mL/min/1.73 m^2^)	177	74.880	28.408	0.991	0.350	58.940	87.348
ΔGFR (worst value) (mL/min/1.73 m^2^)	225	9.713	13.346	0.823	<0.001	3.744	17.090
ΔGFR (worst value) (%)	225	14.135	18.371	0.866	<0.001	5.121	23.492
Start of line 1 to end of follow-Up	225	671.000	676.000	0.753	<0.001	366.000	1042.000
Start of line 2 to end of follow-Up	225	537.000	551.000	0.916	<0.001	299.000	850.000
Recurrence to end of follow-Up	133	189.000	355.000	0.881	<0.001	67.000	422.000

**Table 3 cancers-17-02090-t003:** Univariate analysis of categorical variables compared to AKD flag.

Variable	Chi-Squared (χ^2^)	df	*p*
Sex	3.037	1	0.081
Smoking	2.322	2	0.313
Viral infections	2.427	1	0.119
Diagnosis	25.877	11	**0.007**
Type of monoclonal Ab	12.296	6	0.056
ICI Target	3.858	2	0.145
Line of treatment	12.146	5	**0.033**
Associated chemotherapy	4.7	1	**0.03**
PD-L1 status	0.078	1	0.781
PD-L1 score if pos	2.09	3	0.554
Ulceration	0.0	1	1.0
Regression	0.311	1	0.577
BRAF status	0.434	1	0.51
Vascular invasion	0.549	1	0.459
Lymphatic invasion	0.224	1	0.636
Neural invasion	1.159	1	0.282
TILs	0.384	1	0.536
Baseline LDH	0.077	1	0.781
Race	0.257	1	0.612
Persistent AKD	67.379	1	**<0.001**
eGFR loss >30%	108.614	1	**<0.001**
Toxicity	0.352	1	0.553
Toxicity types	11.513	10	0.319
Toxicity grade	4.473	4	0.346
Toxicity outcome	3.118	2	0.21
Best response	0.322	3	0.956
Progression/recurrence	0.006	1	0.94
Other therapies	0.078	1	0.78
Best response 2	5.59	3	0.133
Death	1.915	1	0.166
Antibiotics	4.443	1	**0.035**
Diabetes	0.788	1	0.375
Hypertension	3.593	1	0.058
Ischemic heart disease	4.869	1	**0.027**
COPD	0.03	1	0.862
CKD	0.974	1	0.324
ACE inhibitors/ARBs	0.783	1	0.376
Calcium channel blockers	1.002	1	0.317
Diuretics	6.054	1	**0.014**
Oral antidiabetics	0.064	1	0.801
Insulin	0.164	1	0.685
Anti-inflammatories	8.155	1	**0.004**
Antiplatelets	0.58	1	0.446
Beta-blockers	1.627	1	0.202
Taxanes	0.358	1	0.55
Platinum	3.757	1	0.053
Targeted therapy	3.123	1	0.077
Immunotherapy	0.777	1	0.378
Chemotherapy	5.865	1	**0.015**
Radiotherapy	0.11	1	0.74

Bold is associated with statistical significance.

**Table 4 cancers-17-02090-t004:** Univariate analysis of continuous variables compared to AKD flag.

	U	*p*-Value
Age (years)	3686.000	0.252
Diagnosis to treatment time (days)	4211.500	0.858
Pack/Year (Tobacco)	2094.000	0.816
Body mass index (BMI)	3596.000	0.204
Body surface area (BSA) (m^2^)	2949.000	**0.006**
WBC (×1.000/mm^3^)	3617.000	0.205
Neutrophils (×1.000/mm^3^)	3740.500	0.340
Lymphocytes (×1.000/mm^3^)	3817.000	0.480
N/L ratio	4305.500	0.590
Eosinophils (×1.000/mm^3^)	3164.500	**0.018**
E/L ratio	3453.000	0.102
Platelets (×1.000/mm^3^)	5082.000	**0.014**
P/L ratio	4771.500	0.084
CRP (basal) (mg/dL)	1421.500	0.865
eGFR (basal) (mL/min/1.73 m^2^)	4389.500	0.490
eGFR after cycle 1 (mL/min/1.73 m^2^)	4974.000	**0.025**
eGFR after cycle 2 (mL/min/1.73 m^2^)	5302.000	**0.003**
eGFR after cycle 3 (mL/min/1.73 m^2^)	4427.000	**0.015**
eGFR after cycle 4 (mL/min/1.73 m^2^)	4505.000	**<0.001**
eGFR after cycle 5 (mL/min/1.73 m^2^)	3867.500	**<0.001**
eGFR after cycle 6 (mL/min/1.73 m^2^)	3572.500	**<0.001**
ΔGFR (worst value) (mL/min/1.73 m^2^)	916.000	**<0.001**
ΔGFR (worst value) (%)	239.000	**<0.001**
Start of line 1 to end of Follow-Up	4009.500	0.918
Start of line 2 to end of Follow-Up	4301.000	0.521
Recurrence to end of Follow-Up	1106.500	0.162

Bold is associated with statistical significance.

**Table 5 cancers-17-02090-t005:** Multivariate logistic regression for predictors of AKD.

Variable	Estimate	Standard Error	Odds Ratio	95% CI Lower	95% CI Upper	*p*-Value
Body surface area (BSA)	2.058	0.923	8.165	0.203	3.997	**0.030**
Anti-inflammatory drugs	3.379	1.379	29.738	0.669	6.096	**0.014**
Antibiotics (Yes)	1.106	0.719	3.022	−0.297	2.509	0.054
Chemotherapy (Yes)	−1.068	0.630	0.343	−3.929	0.258	0.085
Platelet count (Baseline)	−0.004	0.002	0.996	−0.009	0.000	0.061
Eosinophils (Baseline)	0.319	0.206	1.375	−0.132	1.117	0.122
Diuretics (Yes)	1.087	0.595	2.965	−0.080	2.254	0.068

Bold is associated with statistical significance.

## Data Availability

The raw data supporting the conclusions of this article will be made available by the authors on request.

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
