# Peer review of "Immunotherapy-Associated Renal Dysfunction in Metastatic Cancer: An Emerging Challenge in Onco-Nephrology"

_cancers, 2025, doi:10.3390/cancers17132090_

Round 1
Reviewer 1 Report
Comments and Suggestions for Authors
This article presents a well-designed retrospective study evaluating the incidence and predictors of acute kidney disease (AKD) following immune checkpoint inhibitor (ICI) therapy in a real-world cohort of metastatic cancer patients. The article is timely and clinically important because renal complications associated with ICIs are an understudied but growing concern in oncology. However, there are several areas where the study could be improved or further clarified:
1- Please consider at least a brief subgroup analysis or discussion on differential risk by cancer subtype or ICI agent, especially since renal cell carcinoma and melanoma may behave differently.
2- Please include subgroup analyses or survival curves if available, or explain why survival is not affected in some patients despite persistent AKD.
3- Although the authors briefly mention corticosteroids, there is little discussion of clinical management strategies for AKD once detected. Please expand the discussion to address management approaches (e.g., medication retention, nephrology referral, steroids) and how early detection of ACD may alter treatment.
4- Authors should clarify whether any biopsy data were available or whether the diagnosis of ATIN (or other nephropathies) was purely clinical.
5- The study used a single-center retrospective cohort. External validation in a multi-institutional dataset would significantly increase the generalizability of the predictive model.
6- Please ensure that the keywords section (lines 39-40) is appropriately completed.
7- Please avoid unnecessary phrases such as “this is a new concept” that are used multiple times.
8- Were urinalysis data collected? This may help distinguish prerenal ACD from interstitial nephritis.
9- The protective trend of concurrent chemotherapy needs to be better contextualized, as different agents (e.g., platinum and taxane) may have different renal effects.
10- Has any relationship been investigated between ACD and other irAEs? A table comparing patients with and without ACD according to the frequency of other irAEs would add depth.
Reviewer 2 Report
Comments and Suggestions for Authors
Dear Authors,
Thank you for submitting this relevant study on acute kidney disease (AKD) in patients receiving immune checkpoint inhibitors. The work addresses an important gap in onco-nephrology and provides valuable real-world data on an underrecognized complication.
Major Comments
Study population heterogeneity
Your cohort includes multiple tumor types, ICI regimens, and treatment lines without adequate stratification. This limits the ability to identify tumor-specific or ICI-specific risk factors. Consider presenting results by major tumor subtypes (NSCLC, melanoma, RCC) and ICI class (anti-PD1 vs anti-PDL1 vs anti-CTLA4).
Sample size limitations
Several risk factors have very small numbers (e.g., anti-inflammatory drugs n=6), which inflates odds ratios and reduces statistical reliability. The OR of 29.74 for NSAIDs, while biologically plausible, needs interpretation with caution given the small denominator.
Missing histological confirmation
Without renal biopsies, you cannot definitively attribute AKD to ICIs versus other causes (contrast nephropathy, sepsis, drug interactions). This weakens causal inference. At minimum, discuss this limitation and consider a more conservative interpretation of results.
Incomplete long-term follow-up
You report AKD persistence >30 days but don't analyze progression to CKD or dialysis requirements. This information is crucial for clinical decision-making regarding ICI continuation.
Chemotherapy "protective effect"
The trend toward protection with concurrent chemotherapy (OR 0.34, p=0.085) likely reflects selection bias rather than biological effect. Patients receiving combination therapy are typically younger, with better performance status and fewer comorbidities. Address this potential confounding.
Minor Comments
Table 3: Several variables show borderline significance that may not be meaningful. Consider focusing on the most robust predictors.
The analysis of other immune-related adverse events is superficial. A more detailed exploration of irAE correlations would strengthen mechanistic insights.
Consider discussing practical implications: should NSAIDs be contraindicated? What monitoring frequency do you recommend?
Suggestions for Revision
Stratify results by tumor type and ICI class.
Provide more conservative interpretation of risk factors with small numbers.
Expand discussion of study limitations, particularly lack of histological confirmation.
Add analysis of long-term renal outcomes where data permit.
Discuss clinical implications more explicitly.
Additional Suggestions for Added Value
Given the frequency of AKD you report, consider highlighting the need for prospective, possibly multicenter studies to validate these findings, including exploration of non-invasive biomarkers or imaging for improved diagnosis.
You might suggest specific, actionable recommendations for clinicians—such as avoiding NSAIDs when possible, or concrete guidelines for renal monitoring intervals during ICI therapy, especially for high-risk patients.
The predictive model you propose is promising. If validated, it could be further developed into a practical tool or calculator for clinical use.
While I understand that renal biopsies are challenging in this setting, including more detailed clinical or laboratory criteria for diagnosing suspected ICI-related AKD would strengthen future research.
Summary
The core finding that AKD occurs in over 20% of patients is important and supports enhanced renal monitoring in ICI therapy. Your team addresses a critical and often overlooked problem in oncology, and with these revisions, your work will make a solid and clinically relevant contribution to the field. I look forward to seeing this study published after these points are addressed.
Best regards,
Reviewer 3 Report
Comments and Suggestions for Authors
The study uses a retrospective, single-center design. How do the authors account for institutional treatment protocols, referral patterns, or data recording biases that may limit external validity?
Over this 6-year span, clinical guidelines, ICI agents, and diagnostic definitions have evolved. How were these temporal confounders controlled or accounted for?
Why was a 90-day cutoff chosen? Was this based on KDIGO guidance or pragmatic data availability? Would longer follow-up have revealed delayed AKD cases?
Baseline kidney function definition: How reliable is the last pre-treatment serum creatinine value as a reference, especially given possible pre-existing variability in cancer patients?
The confidence interval for BSA (OR 8.17, CI: 1.23–54.0) is extremely wide, suggesting model instability. Could this reflect overfitting or low event-per-variable ratio?
This implausibly high odds ratio suggests either a strong confounder or insufficient control. Were doses, duration, or indication for NSAIDs considered?
Only 133 patients of 226 had CRP values. Were any patients who were missing laboratory data included or was the data imputed? If so, could this missingness impart bias on the study.
With all the univariate testing that was conducted, were any adjustments for multiple comparisons (e.g. Bonferroni, false discovery rate) made to control for false positives?
What proportion of AKD cases would not meet AKI or CKD criteria? Is the added diagnostic resolution of AKD clinically actionable?
How was persistence beyond 30 days assessed in a retrospective design? Were these patients continuously monitored, or could this be an artifact of intermittent follow-up?
The inverse association is both biologically implausible and statistically non-significant (p = 0.085). Should this speculative interpretation be revised or omitted?
Was there any histopathologic confirmation (e.g., kidney biopsy) to support the link between eosinophils and interstitial nephritis? Could eosinophilia reflect systemic inflammation unrelated to renal injury?
The cohort includes lung, renal, melanoma, urothelial, and even breast cancer patients. Were subgroup analyses performed to determine if AKD risk differs by tumor type or ICI class?
Was this variable excluded due to missingness? Could this missingness be non-random, reflecting testing availability or clinical urgency, and therefore be a bias indicator?
The reported changes in eGFR (e.g., ~10 mL/min) are modest. Were these clinically relevant, or possibly attributable to laboratory variability or prerenal factors?
Did the identification of AKD affect any treatment decisions (e.g. ICI discontinuation, steroid therapy)? If not, how do the authors substantiate their suggestion that AKD could be integrated into care pathways?
Are the authors going to validate their predictive model in a prospective cohort, or in other centers to evaluate reproducibility?
Expanding on eosinophils, did the authors have any other potential biomarker (e.g. urinary NGAL, IL-18) that could provide pathophysiological insight?
One author has multiple ties to the pharmaceutical industry. Were any of these ties relevant to ICI-related toxicities, or do authors contend that they would not affect interpretation?
The discussion occasionally overstates conclusions (e.g. "game-changer" or "inevitable recognition"). Would the authors consider softening this term with observational design that generates rather than tests hypotheses?
Round 2
Reviewer 1 Report
Comments and Suggestions for Authors
I am satisfied that the authors have addressed all of my previous concerns about the article. It is now much improved and I feel that it is now suitable for publication.
Reviewer 2 Report
Comments and Suggestions for Authors
Dear Dr. Trevisani and colleagues,
I have carefully reviewed your comprehensive responses to my previous comments and the corresponding revisions to your manuscript. I am pleased to inform you that your responses have fully addressed all of my concerns in a thorough and scientifically rigorous manner.
Summary of Assessment:
Your systematic approach to addressing each reviewer comment demonstrates exceptional scientific integrity and attention to detail. Particularly noteworthy are:
- Transparency and Scientific Honesty: Your explicit acknowledgment of key limitations, including the absence of histological confirmation and small subgroup sizes, enhances the credibility of your work while maintaining its clinical relevance.
- Statistical Rigor: Your cautious interpretation of the NSAID findings (OR 29.74, n=6) exemplifies appropriate statistical reporting and demonstrates responsible data interpretation.
- Clinical Applicability: The addition of practical recommendations regarding NSAID avoidance and monitoring protocols significantly enhances the clinical utility of your research.
- Methodological Clarity: Your detailed explanations of the AKD definition application and study limitations provide readers with the necessary context to properly interpret your findings.
Specific Strengths of Your Responses:
- The addition of text acknowledging clinical-only diagnoses without histological confirmation appropriately sets expectations for readers
- Your balanced discussion of potential selection bias regarding chemotherapy effects shows scientific maturity
- The expanded future research directions demonstrate clear understanding of your study's place within the broader research landscape
- Your practical clinical recommendations bridge the gap between research findings and clinical practice
Scientific Contribution:
Your work makes a valuable contribution to the emerging field of onco-nephrology by:
- Establishing AKD as a clinically significant entity in ICI therapy (20.4% incidence)
- Identifying novel risk factors that can inform clinical practice
- Providing a foundation for future multicenter validation studies
- Offering evidence-based recommendations for clinical monitoring
Final Decision:
Based on your comprehensive responses and the clinical significance of your findings, I am satisfied that this manuscript meets the standards for publication. Your work addresses an important knowledge gap in cancer care and provides actionable insights that will benefit both clinicians and patients.
Recommendation: ACCEPT
The manuscript is ready for publication. Your responses demonstrate that you have thoughtfully considered all feedback and made appropriate revisions that strengthen both the scientific rigor and clinical applicability of your research.
Congratulations on producing a well-executed study that advances our understanding of immune checkpoint inhibitor-associated nephrotoxicity.
Best regards,
Reviewer 3 Report
Comments and Suggestions for Authors
The paper can be accepted in its present form.